# Dissipation, Metabolism, Accumulation, Processing and Risk Assessment of Fluopyram and Trifloxystrobin in Cucumbers and Cowpeas from Cultivation to Consumption

**DOI:** 10.3390/foods12102082

**Published:** 2023-05-22

**Authors:** Kai Cui, Shuai Guan, Jingyun Liang, Liping Fang, Ruiyan Ding, Jian Wang, Teng Li, Zhan Dong, Xiaohu Wu, Yongquan Zheng

**Affiliations:** 1Institute of Quality Standard and Testing Technology for Agro-Products, Shandong Academy of Agricultural Sciences, Shandong Provincial Key Laboratory of Test Technology on Food Quality and Safety, Jinan 250100, China; 2Institute of Plant Protection, Chinese Academy of Agricultural Sciences, Beijing 100193, China

**Keywords:** fluopyram and trifloxystrobin, residue behavior, residue accumulation, food processing, health risk assessment

## Abstract

Fluopyram and trifloxystrobin are widely used for controlling various plant diseases in cucumbers and cowpeas. However, data on residue behaviors in plant cultivation and food processing are currently lacking. Our results showed that cowpeas had higher fluopyram and trifloxystrobin residues (16.48–247.65 μg/kg) than cucumbers (877.37–3576.15 μg/kg). Moreover, fluopyram and trifloxystrobin dissipated faster in cucumbers (half-life range, 2.60–10.66 d) than in cowpeas (10.83–22.36 d). Fluopyram and trifloxystrobin were the main compounds found in field samples, and their metabolites, fluopyram benzamide and trifloxystrobin acid, fluctuated at low residue levels (≤76.17 μg/kg). Repeated spraying resulted in the accumulation of fluopyram, trifloxystrobin, fluopyram benzamide and trifloxystrobin acid in cucumbers and cowpeas. Peeling, washing, stir-frying, boiling and pickling were able to partially or substantially remove fluopyram and trifloxystrobin residues from raw cucumbers and cowpeas (processing factor range, 0.12–0.97); on the contrary, trifloxystrobin acid residues appeared to be concentrated in pickled cucumbers and cowpeas (processing factor range, 1.35–5.41). Chronic and acute risk assessments suggest that the levels of fluopyram and trifloxystrobin in cucumbers and cowpeas were within a safe range based on the field residue data of the present study. The potential hazards of fluopyram and trifloxystrobin should be continuously assessed for their high residue concentrations and potential accumulation effects.

## 1. Introduction

Greenhouse cultivation has gradually developed into the dominant production approach to achieve a year-round supply of various vegetables, including cucumbers and cowpeas. However, these vegetables suffer from serious plant diseases due to their semi-enclosed and comfortable environments [1], resulting in the frequent application of fungicides. Fluopyram, N-{2-[3-chloro-5-(trifluoromethyl)pyridin-2-yl]ethyl}-2-(trifluoromethyl)benzamide, (FLU, Figure 1), a new pyridyl-benzamide fungicide, acts on the enzyme succinate dehydrogenase to inhibit spore germination, mycelium growth and sporulation. Trifloxystrobin, methyl (2E)-(methoxyimino)(2-{[({(1E)-1-[3-(trifluoromethyl)phenyl]ethylidene}amino)oxy]methyl}phenyl)acetate, (TRI, Figure 1), a strobilurin fungicide, acts as a quinone outside inhibitor to inhibit the mitochondrial respiration of pathogens. FLU and TRI are often simultaneously or alternately used to control a variety of plant diseases, such as cucumber and cowpea anthracnose, cucumber powdery mildew and target spot, achieving a high control efficiency due to their strong synergistic effects [2]. However, increasing evidence confirms their potential toxic effects on mammals, such as the induction of thyroid and liver tumor formation by FLU [3,4] and the triggering of neurotoxicity and skin toxicity by TRI [5,6]. Considering the potential health hazards of FLU and TRI, the residue fate and dietary health risks associated with exposure to FLU and TRI should be studied in agricultural products following their application.

Some studies have reported on the residue levels, dissipation behavior and dietary risk assessment of FLU or TRI in different farm crops [7,8,9,10,11,12,13]. However, no work to date has focused on the potential accumulation of pesticide residues in crops after repeated spraying. It is generally recommended that fungicides are sprayed two to three times during crops’ development periods; more alarmingly, farmers often increase their application according to disease severity and their farming experience. This may result in residue accumulation of pesticides in crops and may pose a high dietary risk to humans. Single and repeated applications of fungicides have been proven to generate different pesticide dissipation rates in crops [14]. Hence, understanding the residue dissipation and accumulation of FLU and TRI after repeated spraying is crucial to ensure food safety and protecting public health.

Agricultural products are generally processed before being eaten through operations such as peeling, washing, boiling, stir-frying and pickling. These processing operations play an important role in the reduction or increase in pesticide residues in processed agricultural products [15,16,17,18,19]. To date, only a few papers have been published on the changes in FLU and TRI residues during food processing. Słowik-Borowiec and Szpyrka found that washing, peeling, juicing, boiling and ultrasonic washing could remove large amounts (≥56%) of FLU from apples [20]. On the contrary, TRI residues increased 3.7–5.4 times in dried red peppers following hot-air and sunlight drying [21]. Studies have shown that parent pesticides may be metabolized into more toxic compounds during food processing [22,23,24,25]. Owing to the wide use of FLU and TRI on cucumbers and cowpeas, their main toxic metabolites, i.e., fluopyram benzamide (FLB) and trifloxystrobin acid (TRA) (Figure 1), should also be measured and assessed to achieve a comprehensive risk assessment.

Our study is the first to systematically study the dissipation, metabolism, accumulation, processing and risk assessment of FLU and TRI and their main metabolites (FLB and TRA) in cucumbers and cowpeas from cultivation to consumption. Firstly, greenhouse-field trials were conducted to investigate the dissipation behavior and accumulation potential of pesticides applied to cucumbers and cowpeas. Secondly, a preliminary study of changes in pesticide residues in cucumbers and cowpeas was conducted via several traditional household processing operations. Thirdly, the health risks associated with exposure to pesticides were assessed on the basis of our residue data. The findings of this study could help in understanding the fate of FLU and TRI residues in different scenarios and could provide a reference for the rational use of these pesticides.

## 2. Materials and Methods

### 2.1. Chemicals and Materials

Standards of FLU, FLB, TRI and TRA (all 1000 mg/L) were supplied by the Alta Scientific Co., Ltd. (Tianjin, China). Acetonitrile and formic acid used were of high-performance liquid chromatography (HPLC) grade and were acquired from Macklin Biochemical Co., Ltd. (Shanghai, China). The anhydrous magnesium sulfate (MgSO_4_) and sodium chloride (NaCl) used were of analytical grade and were acquired from Sinopharm Chemical Reagent (Shanghai, China). The sorbents (primary, secondary amine, PSA; graphitized carbon black, GCB) and 0.22 μm nylon syringe filters were purchased from Agela Technologies (Beijing, China). All standard solutions were stored at −20 °C in the dark.

### 2.2. Greenhouse-Field Trials

The greenhouse-field trials were conducted at the experimental base of Shandong Academy of Agricultural Sciences (Jiyang District, Shandong, China) at 36°98′ N and 116°98′ E. No FLU or TRI had been applied to any of the experimental plots for 2 years. Each vegetable variety underwent two treatments, i.e., each had an experiment plot (100 m^2^) and a control plot (50 m^2^). A 43% suspension concentrate of FLU (21.5%) and TRI (21.5%) was sprayed three times every 7 days on cucumbers and cowpeas at the recommended dosage of 375 mL/ha. Three independent fresh samples (~2.0 kg) were randomly collected from each plot at 2 h, 1 d, 3 d, 5 d and 7 d after each application to study the dissipation and accumulation of pesticides. To study the effect of processing on pesticide residues, ≥20.0 kg of samples were collected 1 day after the last application. All processed and unprocessed samples were chopped, homogenized and frozen at −20 °C before analysis.

### 2.3. Processing Operations

The household processing procedure mimicked traditional Chinese cooking. The processing steps were conducted as follows based on modified versions of Huan et al. and An et al. [26,27].

Peeling: 1.0 kg of cucumber samples were peeled. All skins and pulps were stored separately.

Washing: 1.5 kg of samples were immersed in 4.5 L of tap water (25 °C) and then manually washed. Five sampling time points were set: 1 min, 3 min, 5 min, 7 min and 10 min.

Stir-frying: The samples were first chopped into 3–4 cm pieces. Then, 50 mL of peanut oil and 1.5 kg of chopped samples were sequentially added into an electric frying pan. The samples were stir-fried evenly at 1000 W. Five sampling time points were set: 1 min, 3 min, 5 min, 7 min and 10 min.

Boiling: 1.5 kg of chopped samples were immersed in 4.5 L of boiling tap water. Five sampling time points were set: 1 min, 3 min, 5 min, 7 min and 10 min.

Pickling: 1.5 kg of chopped samples were added to a 3.0 L sealed plastic jar, and 2.0 L of 7% NaCl aqueous solutions was then added. All jars were finally stored in an incubator at 25 °C. Six sampling time points were set: 2 h, 1 d, 3 d, 5 d, 7 d and 14 d.

After sampling, the filter papers were used to remove the excess liquids (water or oil). Finally, each sample was homogenized and frozen at −20 °C until analysis.

### 2.4. Analytical Procedure

The homogenized samples (10 ± 0.01 g) were weighed and added to 50 mL polypropylene centrifuge tubes. A total of 10 mL of acetonitrile (acidified with 1% formic acid, *v*/*v*) was added into each tube and mixed thoroughly for 5 min using a multi-tube vortex mixer (2500 r/min). To all tubes, we added 1.5 g of NaCl and 4 g anhydrous MgSO_4_, and the tubes were shaken for an additional 1 min. After centrifuging for 5 min at 5000 r/min, the aliquot of the extract (1.5 mL) was transferred into a 2 mL cleanup tube with the addition of 20 mg GCB, 50 mg PSA and 150 mg anhydrous MgSO_4_. The mixture was then mixed thoroughly for 1 min and centrifuged for 5 min at 5000 r/min. After filtration via a 0.22 µm nylon syringe, the resulting filtrates were prepared for subsequent HPLC-triple-quadrupole mass spectrometer (MS/MS) analysis.

For quantitative analysis of FLU and TRI and their main metabolites (FLB and TRA), a 1290 Infinity II HPLC equipped with a 6495 MS/MS (Agilent, Santa Clara, CA, USA) with a Poroshell 120 EC-C18 column (2.1 mm × 50 mm, i.d., 2.7 μm, Agilent, USA) was used. The HPLC–MS/MS was operated in electrospray-positive ionization (ESI+) and multiple reaction monitoring (MRM) modes. The column temperature and injected volume were set to 40 °C and 2 μL, respectively. The mobile phase was composed of acetonitrile (A) and 0.1% formic acid aqueous solution (B) (0.3 mL/min), with the following parameters: 0 min, 10% A; 0.5 min, 10% A; 2.5 min, 90% A; 3.5 min, 90% A; 3.6 min, 10% A; 5 min, 10% A. The ESI parameters were as follows: capillary voltage, 3.5 kV; nozzle voltage, 500 V; gas temperature, 200 °C; gas flow, 11 L/min; nebulizer, 25 psi; sheath gas temperature, 300 °C; sheath gas flow, 12 L/min. Details of the MRM conditions for the analysis of the four analytes are shown in Appendix A.

### 2.5. Health Risk Assessment

The health risks to consumers associated with exposure to FLU and TRI were assessed based on the risk quotient (RQ) method. The national estimated daily intake (NEDI) and chronic risk quotient (RQc) were calculated to assess chronic dietary intake risk using Equations (1) and (2). Moreover, the international estimation of short-term intake (IESTI) and the acute risk quotient (RQa) were calculated to assess acute dietary intake risk using Equations (3)–(5) [28].
NEDI = STMR × F/bw/1000(1)
RQc = NEDI/ADI(2)
IESTI = LP × HR/bw/1000(3)
IESTI = LP × HR × v/bw/1000(4)
RQa = IESTI/ARfD(5)

In the above equations, STMR and HR represent the supervised trial median residue and the highest residue in greenhouse cowpeas and cucumbers (μg/kg), respectively. F refers to the daily intake of cowpeas and cucumbers (g/d), and bw represents the average body weight of a Chinese child or adult (kg). ADI and ARfD represent the acceptable daily intake and acute reference dose for FLU and TRI (μg/kg bw/d), respectively. LP is the large portion for cowpeas and cucumbers (g/d), and v is the variability factor. When an RQ value is larger than 1, it is considered to pose an unacceptable health risk. A higher RQ value represents a greater health risk. Notably, Equation (3) is used to calculate the IESTI for cowpeas, and Equation (4) is used for cucumbers. All parameters used for the calculation are supplied in Appendix A.

### 2.6. Analysis of Results

The dissipation kinetics for FLU and TRI in cucumbers and cowpeas were calculated using Equation (6), and the half-life (t_1/2_) was calculated using Equation (7) [29].
C_t_ = C_0_ × e^−kt^(6)
t_1/2_ = (ln2)/k(7)

In Equations (6) and (7), C_t_ represents the sample residue at time t (μg/kg), C_0_ represents the initial sample residue (μg/kg), and k refers to the dissipation rate constant.

FLU and TRI residue accumulation (RA) in cucumbers and cowpeas after repeated spraying was calculated using Equation (8) [14].
RA = 1: (C_2_/C_1_): (C_3_/C_2_): …… (C_n_/C_n−1_)(8)

In Equation (8), C_n_ represents the mean residue for a pesticide at each same time point after n instances of repeated spraying (μg/kg).

The processing factor (PF) for FLU and TRI during household processing operations was calculated using Equation (9) [30].
PF = C_pp_/C_rap_(9)

In Equation (9), C_pp_ represents the pesticide residue concentrations in the processed products (μg/kg), and C_rap_ represents the pesticide residue concentrations in the raw agricultural products (μg/kg). PF > 1 represents pesticide residue concentration and PF < 1 represents pesticide residue reduction.

The matrix effect (ME) reflects the effects of co-extractives on the signal increase/decrease in pesticides and was calculated using Equation (10) [31].
ME = slope (matrix)/slope (solvent) × 100%(10)

In Equation (10), slope (matrix) represents the slope of the matrix calibration curve, and slope (solvent) represents the slope of the solvent calibration curve. In general, an |ME| ≤ 10% represents an acceptable matrix effect. An ME < −10% represents a signal suppression effect, and an ME > +10% represents a signal enhancement effect.

The residue definition of FLU was defined as the sum of FLU and its main metabolite, FLB; it is expressed as FLU_sum_ and was calculated using Equation (11). The TRI residue was defined as the sum of TRI and its main metabolite, TRA (expressed as TRI_sum_) and calculated using Equation (12).
FLU_sum_ = C_FLU_ + 1.04 × C_FLB_(11)
TRI_sum_ = C_TRI_ + 2.10 × C_TRA_(12)

In Equations (11) and (12), C_FLU_, C_FLB_, C_TRI_ and C_TRA_ (μg/kg) represent the residue concentrations of FLU, FLB, TRI and TRA. In addition, a 1/2 limit of quantification (LOQ) value is assigned to the residue concentrations that are below the LOQs.

## 3. Results

### 3.1. Method Validation

In our study, the methods for the simultaneous analysis of FLU and TRI and their metabolites, FLB and TRA, were validated in terms of linearity, LOQ, ME, precision and accuracy. Linearity was estimated based on the solvent and matrix standard calibration curves in the range of 1 μg/L to 5000 mg/L, with satisfactory correlation coefficients (R^2^) ranging from 0.9943 to 1. The LOQs, defined as the lowest spiked concentrations in the matrix samples, were 1 μg/kg for the four analytes in the raw and processed cucumbers and cowpeas (Appendix A). The ME values ranged from −79.41% to 16.00% for different matrices (Appendix A), and therefore, matrix standard matrix curves were used to determine the four analytes and eliminate the influence of matrix effects. The precision and accuracy of this method were estimated based on a recovery assay experiment, in which we assessed the terms of the recoveries and the relative standard deviation (RSD). The average recoveries of the four analytes at the four spiked concentrations ranged from 83.73% to 115.78% in different matrices, with RSDs of 0.48–14.38% (Figure 2) (validation criteria: recovery of 70–120%, RSD ≤ 20%). These results confirm that the proposed analytical method was suitable for the extraction and detection of FLU and TRI and their two metabolites in cucumber and cowpea matrices.

### 3.2. Residue Dissipation and Accumulation of FLU and TRI in Greenhouse Cucumbers and Cowpeas

In our study, cucumbers and cowpeas were sprayed with FLU and TRI three consecutive times every seven days; all the residue data and dissipation curves are listed in Table 1 and Figure 3. The FLU residues in cucumbers ranged from 21.44 μg/kg to 107.13 μg/kg following the first spraying event, 120.33 μg/kg to 225.89 μg/kg following the second spraying event and 152.00 μg/kg to 247.65 μg/kg following the third spraying event, and those in cowpeas were 877.37–1341.08 μg/kg, 1626.35–2341.41 μg/kg and 2103.66–3256.29 μg/kg, respectively. These results suggest that cowpeas had much higher levels of FLU residues than cucumbers. A similar residue pattern was found between TRI levels in cucumbers and cowpeas (Table 1). FLU and TRI residues were then used to fit the first-order kinetics models, which produced satisfactory correlation coefficients of 0.50–0.97. The calculated half-lives of FLU were 3.15, 8.89 and 10.66 days for cucumbers and 12.37, 16.91 and 15.40 days for cowpeas, and those for TRI were 2.60, 4.88 and 7.97 days for cucumbers and 17.33, 22.36 and 10.83 days for cowpeas. These results suggest that FLU dissipated faster in cucumbers than in cowpeas, whereas TRI showed a similar dissipation trend in cucumbers and cowpeas. In addition, we noted that residual FLU and TRI in cucumbers and cowpeas increased with increasing spraying times. The average RA values of FLU were 1:2.65:1.08 for cucumbers and 1:1.83:1.15 for cowpeas, while those for TRI were 1:2.04:1.25 for cucumbers and 1:1.69:1.17 for cowpeas.

The parent FLU and TRI were the main compounds present in the greenhouse-field cucumbers and cowpeas, and their metabolites (FLB and TRA) fluctuated at low residue levels. Residual FLB was only found in cowpeas, with detectable concentrations of 1.14–9.06 μg/kg. Residual TRA ranged from 1.69 μg/kg to 24.26 μg/kg in cucumbers and from 5.85 μg/kg to 76.17 μg/kg in cowpeas, indicating higher levels of TRA residues in cowpeas (Table 1). In general, the concentrations of TRA first increased and then gradually decreased 2 h–7 d after each spraying event, except for the first instance of spraying on cowpeas, whereby TRA concentrations underwent a sustained increase (Figure 3). Although the levels of FLB and TRA residues were low, both showed strong residue accumulation in cucumbers (average RA value of TRA, 1:2.71:1.53) and cowpeas (FLB, 1:4.01:1.99; TRA, 1:2.86:1.14). Together, these results verify the residue accumulation of parent FLU and TRI and their metabolites, FLB and TRA, in greenhouse cucumbers and cowpeas after repeated spraying.

### 3.3. Residue Fate and Processing Factors of FLU and TRI in Cucumbers and Cowpeas

Pesticide residue changes in cucumbers and cowpeas were studied during several traditional household processing operations, including peeling, washing, stir-frying, boiling and pickling; all the data are listed in Appendix A. Pesticide residue reduction and concentration during processing were estimated using the PFs, which are shown in Figure 4. Peeling removed 70.06% of FLU, 76.40% of TRI and 86.80% of TRA from cucumbers (Appendix A). Following washing, the PFs in cucumbers decreased from 0.77 (1 min) to 0.38 (10 min) for FLU and 0.90 (1 min) to 0.30 (10 min) for TRI. A similar stepped downward trend was found for PFs in cowpeas for both FLU and TRI following washing (Figure 4A). Following boiling, the PFs of FLU and TRI also presented a decreasing trend in cucumbers and cowpeas with increasing in processing time (Figure 4C). However, following stir-frying, the PFs of FLU increased from 0.29 to 0.58 for cucumbers and 0.61 to 0.74 for cowpeas in 10 min, and those for TRI increased from 0.12 to 0.47 for cucumbers and 0.55 to 0.77 for cowpeas (Figure 4B). Following pickling, the PFs of FLU and TRI first decreased and then gradually increased (Figure 4D), implying that overpickling may result in an increase in pesticide residues in pickled cucumbers and cowpeas. Although the pesticide residues fluctuated greatly during processing, all operations reduced FLU and TRI residues in cucumbers and cowpeas because their PFs were lower than 1.

In addition to the parent FLU and TRI, the residues of FLB and TRA changed greatly in the processed cucumbers and cowpeas (Appendix A). Washing, stir-frying and boiling removed large amounts of TRA from cucumbers and cowpeas with PFs of 0.19–0.90, except for cucumbers washed for 1 min (1.11); on the contrary, pickling increased the concentration of TRA residues in cucumbers (PF range, 1.39–2.73) and cowpeas (PF range, 1.35–5.41). Residual FLB was low in the processed cowpeas at 1.41–6.91 μg/kg, and PF values of higher than 1 were found for stir-frying for 7 min (1.17) and 10 min (1.23) and boiling for 5 min (1.17) and 7 min (1.23). In total, more attention should be paid to pickling operations due to their obvious effects on TRA concentrations during processing.

### 3.4. Dietary Risk Assessment of FLU and TRI in Cucumbers and Cowpeas

The maximum residue limits (MRLs) were first used to estimate the residue risks of FLU and TRI in cucumbers and cowpeas. The total FLU and TRI residues (expressed as FLU_sum_ and TRI_sum_, respectively) at different sampling intervals are listed in Appendix A. A 3-day pre-harvest interval (PHI) for the 43% suspension concentrate of FLU and TRI was proposed for both cucumbers and cowpeas. In cucumbers, even the highest residues of FLU_sum_ (253.21 μg/kg) and TRI_sum_ (277.61 μg/kg) were less than the MRLs recommended by China, the CAC, the US, the EU, Japan and Korea (500–1000 μg/kg for FLU and 300–700 μg/kg for TRI). In cowpeas, the average FLU_sum_ residues 3 days after each spraying event ranged from 1180.10 to 2214.06 μg/kg and were higher than the MRLs of China and CAC (1000 μg/kg) but lower than the MRLs of the US and the EU (3000–4000 μg/kg). Moreover, the average TRI_sum_ residue 3 days after the first spraying event (1251.89 μg/kg) was lower than the MRLs of the US and the EU (1500 μg/kg), whereas those following the second and third spraying events (2245.24 μg/kg and 2479.15 μg/kg) were higher than these MRLs.

The chronic and acute risks associated with exposure to FLU and TRI from cucumbers and cowpeas were estimated based on FAO/WHO models; the risk values are listed in Table 2. For the chronic risk assessment, the HQc values of FLU for children and adults ranging from 0.0033 to 0.042 for cucumbers and 0.016 to 0.099 for cowpeas, and those for TRI were 0.0010–0.011 for cucumbers and 0.016–0.099 for cowpeas, which were all far below 1. For the acute risk assessment, the HQa values ranged from 0.0027 to 0.017 for cucumbers and 0.023 to 0.057 for cowpeas for both children and adults, i.e., <1. In addition, acute risk assessment for TRI was not considered as the ARfD value was reported to be unnecessary by JMPR and EFSA [32,33]. In conclusion, the human health risks of dietary exposure to FLU and TRI from cucumber and cowpea consumption were considered to be acceptable based on the results of the present study.

## 4. Discussion

Studying the dissipation, metabolism and accumulation of pesticides after repeated spraying is of vital importance to ensure food safety. In our study, FLU and TRI residues were higher in cowpeas than in cucumbers. The residue difference between cucumbers and cowpeas may be largely due to differences in the crops’ morphological characteristics, including crop height, foliar size and shape, fruit size, shape and surface [34]. Moreover, FLU and TRI residues dissipated faster in cucumbers (half-life range, 2.60–10.66 days) than in cowpeas (half-life range, 10.83–22.36 days). During the experimental period, cucumbers grew rapidly to contribute to the dilution of pesticide residues in fruits; however, cowpeas only showed a slight increase in fruit weight after pesticide spraying. This may be the main reason for the fast dissipation of FLU and TRI in cucumbers. Moreover, the half-lives of cowpeas and cucumbers in our study were larger than those in the chili (1.23–2.38 for FLU; 1.27–4.88 for TRI) and onion (1.74–1.83 for FLU; 4.73–4.78 for TRI) [9,12]. Our results suggest that both FLU and TRI showed potential residue accumulation in cucumbers (average RA value, 1:2.65:1.08 for FLU and 1:2.04:1.25 for TRI) and cowpeas (average RA value, 1:1.83:1.15 for FLU and 1:1.69:1.17 for TRI) after repeated spraying. Similarly, Wang et al. confirmed the residue accumulation of four common fungicides in strawberries in the order of procymidone > cyprodinil > pyrimethanil > pyraclostrobin after repeated spraying [14]. In addition to the parent compounds, their metabolites, FLB and TRA, also accumulated in cucumbers (average RA values of TRA, 1:2.71:1.53) and cowpeas (FLB, 1:4.01:1.99; TRA, 1:2.86:1.14). Other studies have also confirmed the prevalence of FLB and TRA in crops [12,13,35]. Therefore, due attention should be given to these metabolites to achieve a comprehensive risk assessment of FLU and TRI in cucumbers and cowpeas.

Food processing is typically accompanied by pesticide residue changes. In our study, peeling, washing, stir-frying, boiling and pickling partially or substantially removed FLU and TRI residues from raw agricultural products (PF range, 0.12–0.97). Unlike washing and boiling operations, residues of FLU and TRI presented an increasing trend with increasing stir-frying and pickling time. We inferred that the main reason for this residue increase was the water loss that occurs in cucumbers and cowpeas under high-temperature conditions (stir-frying) and in high-salt environments (pickling). Similarly, Lu et al. found that residues of organophosphorus pesticides, such as chlorpyrifos, isocarbophos, profenofos and triazophos, were also increased to some extent during the cabbage pickling process with different NaCl solutions [36]. Huan et al. found that residual procymidone, chlorothalonil and difenoconazole increased 1.03–1.11 times in stir-fried cowpeas [27]. More notably, residues of TRA appeared to be concentrated in pickled cucumbers and cowpeas (PF range, 1.35–5.41), and these concentration effects increased with increasing pickling time. TRI was easily hydrolyzed into TRA via cleavage of its methyl ester group in aquatic environments [37,38].

The health risks of chronic and acute exposure to total FLU and TRI from cucumbers and cowpeas were assessed for both children and adults. The HQc and HQa values of total FLU and TRI ranged from 0.0010 to 0.099 and 0.0027 to 0.057, respectively, although the highest value was still far smaller than 1, indicating low risks of adverse effects following exposure to FLU and TRI based on the results of the present study. Similarly, previous studies also revealed a low dietary risk of exposure to FLU or TRI from various agricultural products, such as onions, carrots, strawberries and mangoes [12,39,40,41]. The risks associated with exposure to FLU and TRI should not be ignored. On the one hand, FLU and TRI residue concentrations were high in cucumbers and cowpeas, and some residues at the PHI even exceeded the MRLs established by the Chinese government. On the other hand, long-term exposure to low doses of pesticides (even at permitted levels) could have harmful effects on mammals [42,43,44]. Moreover, numerous studies have confirmed the cumulative toxicity of different pesticides that were classified in both the same and different classes [31,45,46]. Therefore, individual or mixed processing operations should be used to reduce FLU and TRI residues in agricultural products, with the aim of protecting human health.

## 5. Conclusions

In this study, we comprehensively assessed the dissipation, metabolism, accumulation, processing and risk assessment of FLU and TRI in cucumbers and cowpeas from cultivation to consumption for the first time. Our results suggest that FLU and TRI residues were higher in cowpeas but dissipated faster in cucumbers. FLU and TRI were the main compounds found in field samples (≤3256.29 μg/kg), and their metabolites (FLB and TRA) fluctuated at low residue levels (≤76.17 μg/kg). Moreover, both FLU and TRI accumulated in cucumbers and cowpeas after repeated spraying. Our results show that peeling, washing, stir-frying, boiling and pickling could partially or substantially remove FLU and TRI residues from raw cucumbers and cowpeas; on the contrary, residues of the metabolite TRA were obviously concentrated following pickling. Chronic and acute risk assessments indicate that exposure to FLU and TRI through cucumber and cowpea consumption posed a low health risk to either children or adults based on the results of the present study. In the future, more experimental sites and crop categories across China should be chosen to study the fate of FLU and TRI residues to achieve a comprehensive understanding of their actual dietary risks; this would be a significant step in ensuring the safe use of FLU- and TRI-containing products and in protecting human health.

## Figures and Tables

**Figure 1 foods-12-02082-f001:**
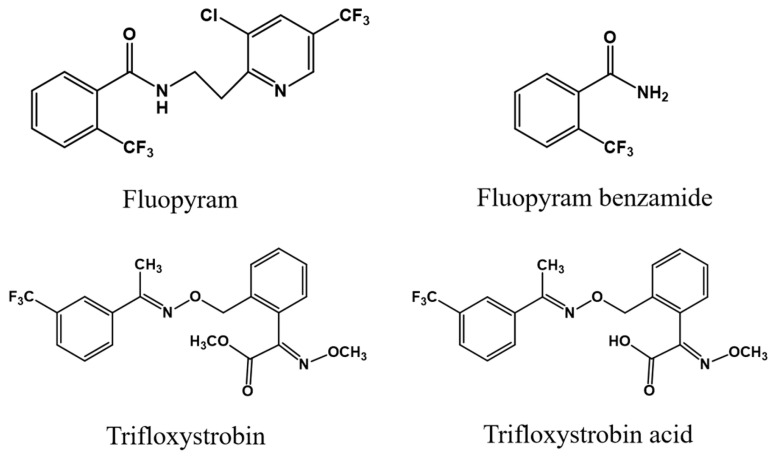
The chemical structures of FLU and TRI and their metabolites, FLB and TRA.

**Figure 2 foods-12-02082-f002:**
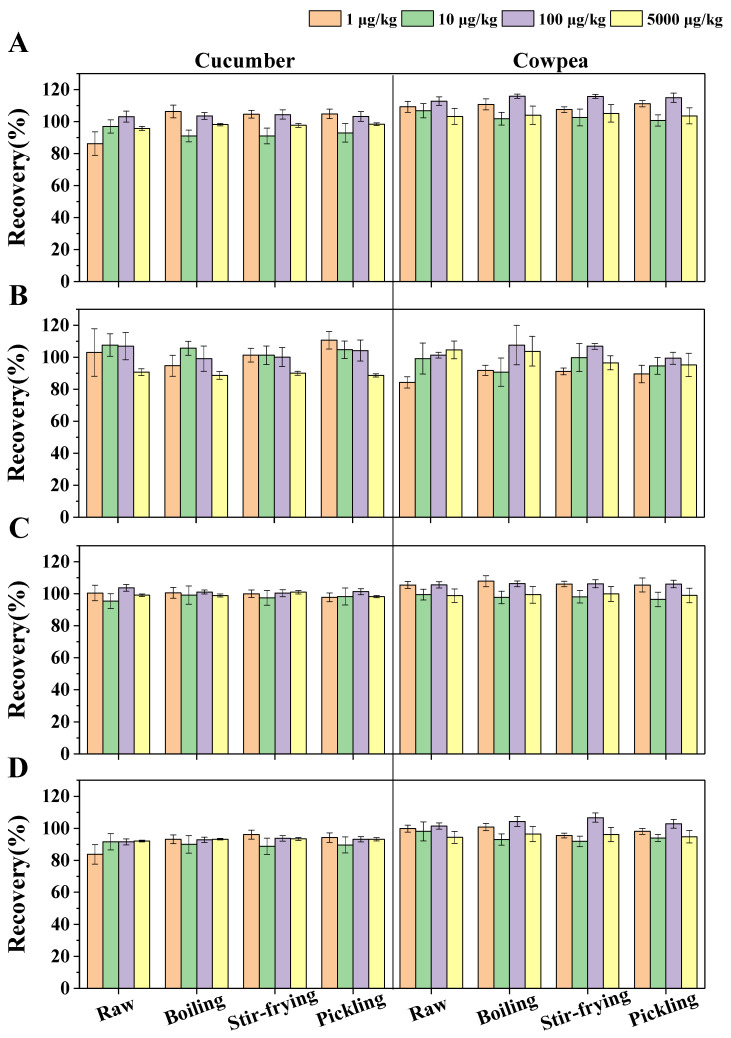
Recoveries of FLU (**A**) and TRI (**B**) and their metabolites, FLB (**C**) and TRA (**D**), in different matrices. Note: The error bars represent standard deviations calculated from five duplicates.

**Figure 3 foods-12-02082-f003:**
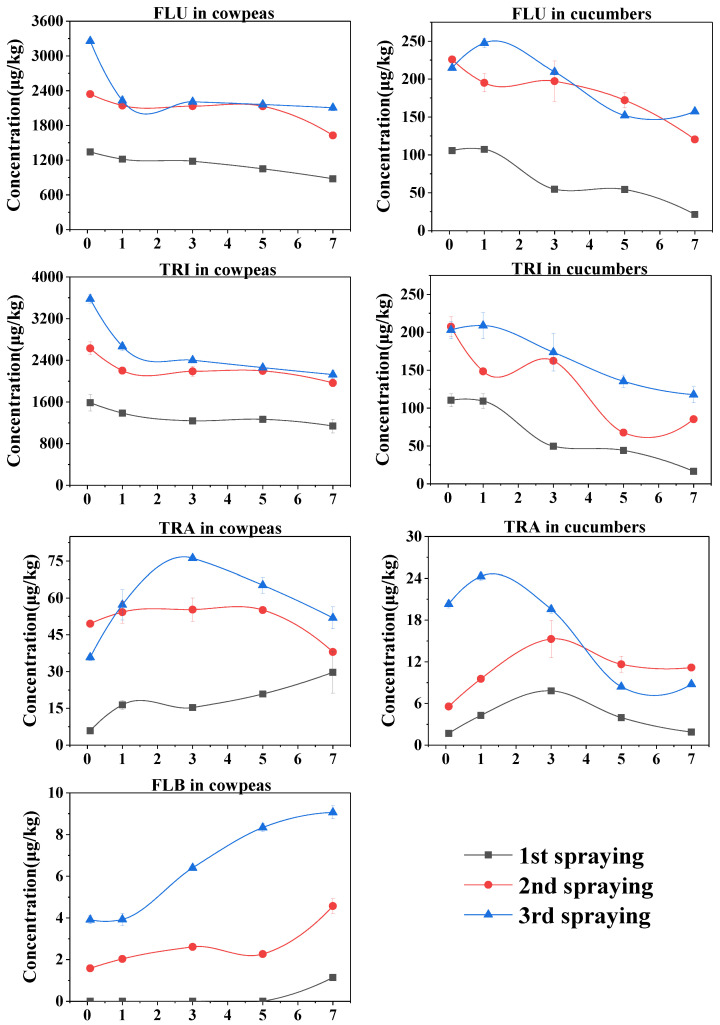
Residue dissipation of FLU and TRI and their metabolites, FLB and TRA, in cucumbers and cowpeas after repeated spraying. Note: The error bars represent standard deviations calculated from three duplicates.

**Figure 4 foods-12-02082-f004:**
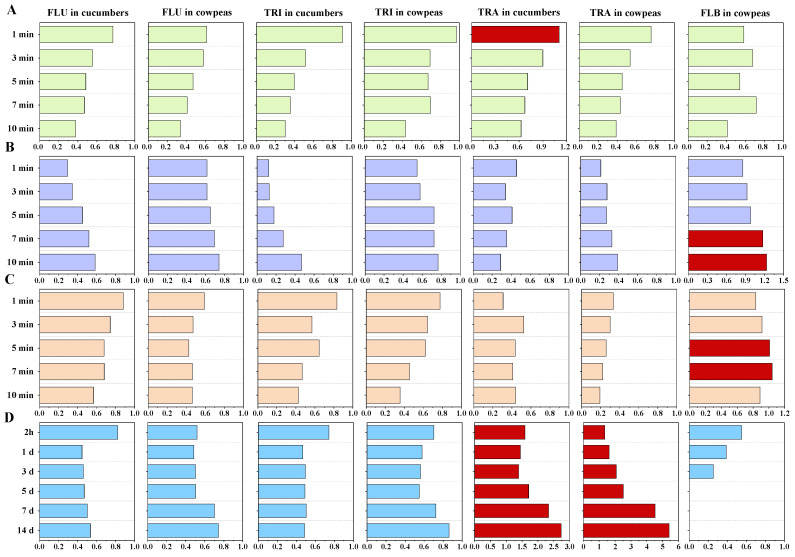
Processing factors (PFs) of FLU and TRI and their metabolites, FLB and TRA, in cucumbers and cowpeas following washing (**A**), stir-frying (**B**), boiling (**C**) and pickling (**D**). The red bars represent a PF value of >1.

**Table 1 foods-12-02082-t001:** Residues, dissipation kinetics and residue accumulation (RA) of FLU and TRI and their metabolites, FLB and TRA, in cucumbers and cowpeas after repeated spraying. Note: SD denotes standard deviation, calculated from three duplicates.

Time	1st Spraying		2nd Spraying		3rd Spraying		RA	Average RA
	Mean Residue ± SD (μg/kg)	Dissipation Rate (%)	Mean Residue ± SD (μg/kg)	Dissipation Rate (%)	Mean Residue ± SD (μg/kg)	Dissipation Rate (%)		
**FLU in cucumbers**								
2 h	105.63 ± 2.14	-	225.89 ± 3.59	-	214.67 ± 2.73	-	1:2.14:0.95	1:2.65:1.08
1 d	107.13 ± 2.68	-	195.11 ± 12.11	13.63	247.65 ± 6.16	-	1:1.82:1.27	
3 d	54.78 ± 2.03	48.14	197.11 ± 26.85	12.74	209.42 ± 4.04	2.44	1:3.60:1.06	
5 d	54.27 ± 1.10	48.62	172.08 ± 9.93	23.82	152.00 ± 2.25	29.19	1:3.17:0.88	
7 d	21.44 ± 0.62	79.70	120.33 ± 1.03	46.73	157.22 ± 2.77	26.76	1:5.61:1.31	
Dynamic equation	y = 120.05e^−0.22x^		y = 228.98e^−0.078x^		y = 237.76e^−0.065x^			
R^2^	0.90		0.85		0.77			
t1/2 (d)	3.15		8.89		10.66			
**FLU in cowpeas**								
2 h	1341.08 ± 53.95	-	2341.41 ± 18.12	-	3256.29 ± 19.45	-	1:1.75:1.39	1:1.83:1.15
1 d	1215.85 ± 27.27	9.34	2142.03 ± 38.70	8.52	2228.24 ± 79.55	31.57	1:1.76:1.04	
3 d	1179.58 ± 39.45	12.04	2131.32 ± 29.49	8.97	2207.41 ± 37.26	32.21	1:1.81:1.04	
5 d	1049.90 ± 5.85	21.71	2130.55 ± 41.46	9.01	2160.21 ± 30.58	33.66	1:2.03:1.01	
7 d	877.37 ± 28.36	34.58	1626.35 ± 7.56	30.54	2103.66 ± 49.25	35.40	1:1.85:1.29	
Dynamic equation	y = 1341.06e^−0.056x^		y = 2351.11e^−0.041x^		y = 2726.41e^−0.045x^			
R^2^	0.95		0.72		0.50			
t1/2	12.37		16.91		15.40			
**TRI in cucumbers**								
2 h	110.14 ± 8.39	-	207.41 ± 13.18	-	202.85 ± 11.54	-	1:1.88:0.98	1:2.04:1.25
1 d	109.14 ± 9.53	0.91	148.30 ± 1.86	28.50	208.68 ± 17.17	-	1:1.36:1.41	
3 d	49.54 ± 2.92	55.02	162.21 ± 4.13	21.79	173.55 ± 24.85	14.45	1:3.27:1.07	
5 d	44.04 ± 1.13	60.02	67.62 ± 0.19	67.40	135.30 ± 8.18	33.30	1:1.54:2.00	
7 d	16.48 ± 0.97	85.03	85.18 ± 4.68	58.93	117.59 ± 10.56	42.03	1:5.17:1.38	
Dynamic equation	y = 126.06e^−0.28x^		y = 194.74e^−0.14x^		y = 216.45e^−0.087x^			
R^2^	0.94		0.74		0.97			
t1/2 (d)	2.60		4.88		7.97			
**TRI in cowpeas**								
2 h	1586.16 ± 156.66	-	2627.72 ± 124.51	-	3576.15 ± 79.16	-	1:1.66:1.36	1:1.69:1.17
1 d	1384.96 ± 24.75	12.68	2200.96 ± 52.53	16.24	2665.77 ± 80.46	25.46	1:1.59:1.21	
3 d	1235.94 ± 29.10	22.08	2187.77 ± 99.08	16.74	2399.93 ± 48.44	32.89	1:1.77:1.1	
5 d	1267.11 ± 60.87	20.11	2194.10 ± 49.65	16.50	2258.87 ± 33.36	36.84	1:1.73:1.03	
7 d	1138.03 ± 131.06	28.25	1964.60 ± 75.54	25.24	2122.85 ± 79.55	40.64	1:1.73:1.08	
Dynamic equation	y = 1495.90 e^−0.040x^		y = 2455.67e^−0.031x^		y = 3140.45e^−0.064x^			
R^2^	0.82		0.69		0.78			
t1/2	17.33		22.36		10.83			
**FLB in cowpeas**								
2 h	<LOQ		1.58 ± 0.04		3.91 ± 0.19		-	1:4.01:1.99
1 d	<LOQ		2.03 ± 0.07		3.93 ± 0.30		-	
3 d	<LOQ		2.61 ± 0.14		6.40 ± 0.09		-	
5 d	<LOQ		2.27 ± 0.08		8.33 ± 0.18		-	
7 d	1.14 ± 0.14		4.57 ± 0.35		9.06 ± 0.31		1:4.01:1.99	
**TRA in cucumbers**								
2 h	1.69 ± 0.11		5.57 ± 0.09		20.30± 0.57		1:3.29:3.64	1:2.71:1.53
1 d	4.27 ± 0.01		9.53 ± 0.32		24.26 ± 0.57		1:2.23:2.55	
3 d	7.81 ± 0.27		15.27 ± 2.69		19.54 ± 0.54		1:1.96:1.28	
5 d	3.96 ± 0.07		11.64 ± 1.15		8.40 ± 0.17		1:2.94:0.72	
7 d	1.89 ± 0.15		11.16 ± 0.06		8.76 ± 0.11		1:5.92:0.79	
**TRA in cowpeas**								
2 h	5.85 ± 0.33		49.52 ± 1.48		35.79 ± 1.22		1:8.47:0.72	1:2.86:1.14
1 d	16.42 ± 1.78		54.21 ± 4.50		57.19 ± 6.12		1:3.30:1.05	
3 d	15.34 ± 0.28		55.26 ± 4.77		76.17 ± 0.68		1:4.97:0.73	
5 d	20.85 ± 0.30		55.05 ± 0.84		65.16 ± 3.18		1:2.64:1.18	
7 d	29.71 ± 8.50		37.98 ± 0.75		51.89 ± 4.48		1:1.28:1.37	

**Table 2 foods-12-02082-t002:** Chronic and acute hazard quotients of FLU and TRI in cucumbers and cowpeas for children and adults.

Pesticides	Vegetables	Spraying Time	NEDI (μg/kg bw/d)		HQc		IESTI (μg/kg bw/d)		HQa	
			Children	Adults	Children	Adults	Children	Adults	Children	Adults
FLU	Cucumber	1	0.11	0.033	0.011	0.0033	2.26	1.37	0.0045	0.0027
		2	0.39	0.12	0.039	0.012	8.72	5.29	0.017	0.011
		3	0.42	0.13	0.042	0.013	8.46	5.14	0.017	0.010
	Cowpea	1	0.52	0.16	0.052	0.016	15.50	11.73	0.031	0.023
		2	0.95	0.29	0.095	0.029	27.41	20.75	0.055	0.041
		3	0.99	0.30	0.099	0.030	28.33	21.45	0.057	0.043
TRI	Cucumber	1	0.13	0.040	0.0033	0.0010	2.73	1.66	-	-
		2	0.38	0.12	0.010	0.0029	8.06	4.90	-	-
		3	0.43	0.13	0.011	0.0032	9.39	5.70	-	-
	Cowpea	1	0.57	0.17	0.014	0.0043	16.02	12.13	-	-
		2	1.00	0.30	0.025	0.0076	29.81	22.56	-	-
		3	1.11	0.34	0.028	0.0084	31.77	24.05	-	-

## Data Availability

The data presented in this study are available on request from the corresponding author.

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
