# Peer review of "Dissipation, Metabolism, Accumulation, Processing and Risk Assessment of Fluopyram and Trifloxystrobin in Cucumbers and Cowpeas from Cultivation to Consumption"

_foods, 2023, doi:10.3390/foods12102082_

Round 1

Reviewer 1 Report

When studies on dissipation, and metabolism are already done for Flu and Tri pesticides as mentioned in the introduction, then why the work was carried out with the title giving the same information?

Secondly, there is no justification for greenhouse trials and no comparison to field trials.

Author Response

Reviewer #1:

Thank you very much for your comments and advice concerning our manuscript. Those comments are all valuable and helpful for revising and improving our manuscript. Based on these comments and suggestions, we have made careful modifications on the revised manuscript. If it still has any problem, please do not hesitate to contact us, and we will try our best to polish the manuscript.

  1. When studies on dissipation, and metabolism are already done for Flu and Tri pesticides as mentioned in the introduction, then why the work was carried out with the title giving the same information?

Response: Thanks for your good question. As you said, several studies have reported the dissipation and metabolism of fluopyram and trifloxystrobin in crops [1-7]. However, the dissipation and metabolism of fluopyram and trifloxystrobin in cowpeas and cucumbers were not included. Factors that influence the persistence of pesticide residues in crops vary, including pesticide varieties, plant species, climate conditions, cultivation patterns, and frequency of application, among other factors. Residue levels and dissipation rates of fluopyram and trifloxystrobin in cowpeas and cucumbers may be different from other crops, thereby producing different residue amounts of metabolites and dietary risks. To achieve a comprehensive evaluation of the potential health risks, the dissipation and metabolism of fluopyram and trifloxystrobin should be comprehensively studied following their application. To the best of our knowledge, this study was the first to systematically assess the dissipation, metabolism, accumulation, processing and risk assessment of fluopyram and trifloxystrobin in cucumbers and cowpeas from cultivation to consumption. Moreover, the metabolism of fluopyram and trifloxystrobin through different processing operations such as peeling, washing, boiling, stir-frying and pickling were also not studied. The findings of this study could help in understanding the fate of fluopyram and trifloxystrobin residues in different scenarios, and could provide a reference for the rational use of these pesticides. Thanks again for your important comments.

References

[1] Cao, M.; Li, S.; Wang, Q.; Wei, P.; Liu, Y.; Zhu, G.; Wang, M. Track of fate and primary metabolism of trifloxystrobin in rice paddy ecosystem. Sci. Total Environ. 2015, 518, 417-423.

[2] Chawla, S.; Patel, D. J.; Patel, S. H.; Kalasariya, R. L.; Shah, P. G. Behaviour and risk assessment of fluopyram and its metabolite in cucumber (Cucumis sativus) fruit and in soil. Environ. Sci. Pollut. Res. 2018, 25, 11626-11634.

[3] Mandal, K.; Singh, R.; Sharma, S.; Kataria, D. Dissipation and kinetic studies of fluopyram and trifloxystrobin in chilli. J. Food Compost. Anal. 2023, 115, 105008.

[4] Matadha, N. Y.; Mohapatra, S.; Siddamallaiah, L. Distribution of fluopyram and tebuconazole in pomegranate tissues and their risk assessment. Food Chem. 2021, 358, 129909.

[5] Mohapatra, S. Residue levels and dissipation behaviors for trifloxystrobin and tebuconazole in mango fruit and soil. Environ. Monit. Assess. 2015, 187, 1-10.

[6] Sharma, N.; Mandal, K.; Sharma, S. Dissipation and risk assessment of fluopyram and trifloxystrobin on onion by GC–MS/MS. Environ. Sci. Pollut. Res. 2022, 29, 80612-80623.

[7] Wang, L.; Li, W.; Li, P.; Li, M.; Chen, S.; Han, L. Residues and dissipation of trifloxystrobin and its metabolite in tomatoes and soil. Environ. Monit. Assess. 2014, 186, 7793-7799.

  1. Secondly, there is no justification for greenhouse trials and no comparison to field trials.

Response: Special thanks for your good question. As you suggested, it is meaningful to consider both greenhouse and open-field trials to comprehensively understand residue behaviour and dietary risks for a pesticide. In China, fluopyram and trifloxystrobin are commonly used fungicides on greenhouse vegetables, including cucumbers and cowpeas. We mainly aimed to investigate the residue behavior of fluopyram and trifloxystrobin in cucumbers and cowpeas under greenhouse environments. Compared with field environments, greenhouse environments are more stable and controllable, which is suitable to study the residual behavior of pesticides. As we konow, climate conditions are important factors to influence pesticide residues in vegetables, including air temperature, precipitation, sunshine hours, cloud cover, relative humidity and wind speed. Generally, higher concentrations of pesticides were observed in greenhouse vegetables than those in open-field vegetables [1-2]. Therefore, our results provide a reference for guiding the rational use of fluopyram and trifloxystrobin in the open-fields on protecting the residents from their potential health hazards. We acknowledge that your questions are very valuable and helpful for our follow-up study, as well as important guiding significance to our future researches. Thanks again for your important comments.

References

[1] Li, Z.; Sun, J.; Zhu, L. Organophosphorus pesticides in greenhouse and open-field soils across China: Distribution characteristic, polluted pathway and health risk. Sci. Total Environ. 2021, 765, 142757.

[2] Yang, G.; Li, J.; Lan, T.; Dou, L.; Zhang, K. Dissipation, residue, stereoselectivity and dietary risk assessment of penthiopyrad and metabolite PAM on cucumber and tomato in greenhouse and field. Food Chem. 2022, 387, 132875.

Reviewer 2 Report

The manuscript describes using FLU and TRI in cucumber and cowpea, dissipation, and accumulation of the agrochemicals. The manuscript is well conducted, the methodology is adequate and the results should increase the quality of the analytical process. However, some of the authors' propositions regarding using FLU and TRI must be reformulated. Isolated studies like this one should not be used as an answer to change substantially the agrochemical policy, as proposed. To this end, systematic and numerous studies must be conducted, therefore the discussions and conclusions must be completely reformulated. For this purpose, I suggest that the manuscript needs major revisions.

In the abstract, abbreviations must be avoided.

The introduction is concise and well-referenced in parts. On p. 3 l.85-91 objectives 2 and 3 proposed are difficult to reach with an isolated study and must be reformulated.

On the text, p. 1-2, please insert before the abbreviation the name of the chemicals.

Methodology brings to the readers elements to be repeated and is fundament in the literature.

On methodology, p. 3 please insert on l.110 the number of samples obtained. In this case, replicates could validate the analytical process.

On the same page, l.114, the authors explain the processing operations, however no references fundament the process. Please insert proper references.

On results p. 7 l.248, the authors described “The results suggested that both FLU and TRI dissipated faster in cucumbers than in cowpeas”. The phrase is not clear, please increase the quality of the discussion in the manuscript using the literature.

Figure 3 and Table 1 are very good.

In the discussion, the authors propose the use of FLU and TRI does not affect substantially human and children's health. Despite the results of the study being concise, I strongly suggest the authors remove such sentences because more studies and deep discussions must be evaluated. The discussion and conclusions must be reformulated in this sense. Phrases such as p.13 l.407-410 must be removed.

Author Response

Reviewer #2:

The manuscript describes using FLU and TRI in cucumber and cowpea, dissipation, and accumulation of the agrochemicals. The manuscript is well conducted, the methodology is adequate and the results should increase the quality of the analytical process. However, some of the authors' propositions regarding using FLU and TRI must be reformulated. Isolated studies like this one should not be used as an answer to change substantially the agrochemical policy, as proposed. To this end, systematic and numerous studies must be conducted, therefore the discussions and conclusions must be completely reformulated. For this purpose, I suggest that the manuscript needs major revisions.

Response: Thank you very much for your comments and advice concerning our manuscript. Those comments are all valuable and helpful for revising and improving our manuscript. Based on these comments and suggestions, we have made careful modifications on the revised manuscript. If it still has any problem, please do not hesitate to contact us, and we will try our best to polish the manuscript. 

We also acknowledge that isolated studies like our study should not be used as an answer to change substantially the agrochemical policy. We have reformulated the propositions regarding using fluopyram and trifloxystrobin. Moreover, we have tried to reformulate the discussion and conclusions. Thanks again for your good suggestion.

  1. In the abstract, abbreviations must be avoided.

Response: Kindly thanks for your reminding. We have deleted all abbreviations in the abstract and used their full name. Please refer to the abstract in the revised manuscript.

  1. The introduction is concise and well-referenced in parts. On p. 3 l.85-91 objectives 2 and 3 proposed are difficult to reach with an isolated study and must be reformulated.

Response: Special thanks for your good suggestion. We have reformulated objectives 2 and 3 for our study. Please refer to Line 87-91 in the revised manuscript.

  1. On the text, p. 1-2, please insert before the abbreviation the name of the chemicals.

Response: Kindly thanks for your reminding. We have added the name of the chemicals before the abbreviation. Please refer to Line 40-42, 44-46 in the revised manuscript.

  1. Methodology brings to the readers elements to be repeated and is fundament in the literature.On methodology, p. 3 please insert on l.110 the number of samples obtained. In this case, replicates could validate the analytical process.

Response: Thanks for the reviewer’s good question. In our study, three independent samples were collected at each sampling time. We have supplemented this part, and please refer to Line 113 in the revised manuscript.

  1. On the same page, l.114, the authors explain the processing operations, however no references fundament the process. Please insert proper references.

Response: Kindly thanks for your reminding. We have added references in the revised manuscript. Please refer to Line 119–121 and References 26, 27 in the revised manuscript.

  1. On results p. 7 l.248, the authors described “The results suggested that both FLU and TRI dissipated faster in cucumbers than in cowpeas”. The phrase is not clear, please increase the quality of the discussion in the manuscript using the literature.

Response: Thanks for your good suggestion. The results suggest that FLU dissipated faster in cucumbers than in cowpeas, whereas TRI showed a similar dissipation trend in cucumbers and cowpeas. We have made the corresponding revision in the revised manuscript. Please refer to Line 257–259 in the revised manuscript.

  1. Figure 3 and Table 1 are very good.

Response: Thank you for your recognition of my work. We will keep working to improve our manuscript.

  1. In the discussion, the authors propose the use of FLU and TRI does not affect substantially human and children's health. Despite the results of the study being concise, I strongly suggest the authors remove such sentences because more studies and deep discussions must be evaluated. The discussion and conclusions must be reformulated in this sense. Phrases such as p.13 l.407-410 must be removed.

Response: Special thanks for your good suggestion. As you said, isolated studies like our study should not be used as an answer to change substantially the agrochemical policy. In our study, the HQc values and HQa values of total fluopyram and trifloxystrobin ranged from 0.0010 to 0.099 and 0.0027 to 0.057, respectively, although the highest value was still far smaller than 1, indicating no risks of adverse effects following exposure to fluopyram and trifloxystrobin. In order to ensure the accuracy of the conclusion, we added “based on the results of the present study” in the discussion and conclusions. We also have reformulated the propositions regarding using fluopyram and trifloxystrobin. Moreover, we have tried to reformulate the discussion and conclusions. Please refer to Discussion and Conclusions in the revised manuscript. Thanks again for your good suggestion.

Reviewer 3 Report

A manuscript entitled "Dissipation, metabolism, accumulation, processing and risk assessment of fluopyram and trifloxystrobin in cucumbers and cowpeas from cultivation to consumption" submitted to Foods presented the dissipation, accumulation, processing and risk assessment of fluopyram and trifloxystrobin and their main metabolites in cucumbers and cowpeas. The topic is important from the point of view of cultivation and obtaining good quality food as well as food safety for humans. The purpose of the work is clearly defined, while the manuscript is properly structured and well written. The research procedures are very well and accurately described but there is a lack of some aspects which should be corrected or supplemented. Detailed comments are listed below:

1.       Please adjust citation to journal requirements

2.       In the introduction section, please highlight a new aspect of the problem addressed in the manuscript.

3.   Section 2.6. “Statistical analysis is not correct, because it is concerned with the analysis of results and not the statistical analysis. Please retitle the section to "Analysis of results" and add the statistical methods used such as standard deviation etc.

4.Table 1. is not very readable and needs to be rewritten or reoriented to horizontal.

5. Figure 2 and 3: Please explain in the figure caption what the error bars mean (It is standard deviation?)

6.    In my opinion, the discussion part of the manuscript should be expanded. The obtained results should be discussed more with the results of other scientists.

7. Conclusion: Please indicate whether the research objective was achieved or the research question was answered. Please suggest also what still needs to be studied in the future and emphasize why the research conducted is important compared to others.

Author Response

Reviewer #3:

A manuscript entitled "Dissipation, metabolism, accumulation, processing and risk assessment of fluopyram and trifloxystrobin in cucumbers and cowpeas from cultivation to consumption" submitted to Foods presented the dissipation, accumulation, processing and risk assessment of fluopyram and trifloxystrobin and their main metabolites in cucumbers and cowpeas. The topic is important from the point of view of cultivation and obtaining good quality food as well as food safety for humans. The purpose of the work is clearly defined, while the manuscript is properly structured and well written. The research procedures are very well and accurately described but there is a lack of some aspects which should be corrected or supplemented. Detailed comments are listed below:

Response: Thank you very much for your comments and advice concerning our manuscript. Those comments are all valuable and helpful for revising and improving our manuscript. Based on these comments and suggestions, we have made careful modifications on the revised manuscript. If it still has any problem, please do not hesitate to contact us, and we will try our best to polish the manuscript.

  1. Please adjust citation to journal requirements.

Response: Kindly thanks for your reminding. We have checked all references and adjusted all references to journal requirements. Please refer to References in the revised manuscript.

  1. In the introduction section, please highlight a new aspect of the problem addressed in the manuscript.

Response: Thanks for your good suggestion. To the best of our knowledge, this study constituted the first to systematically assess the dissipation, metabolism, accumulation, processing and risk assessment of fluopyram and trifloxystrobin in cucumbers and cowpeas from cultivation to consumption. In this study, we focused on studying the potential accumulation of fluopyram and trifloxystrobin in cucumbers and cowpeas after repeated spraying. Moreover, their main metabolites were also detected. As you suggested, we have added some sentences in introduction to highlight a new aspect of the problem addressed in the manuscript in the revised manuscript. Please refer to Line 67–69, 79–82, 83–85, 91–93 in the revised manuscript. Thanks again for your good suggestion.

  1. Section 2.6. “Statistical analysis is not correct, because it is concerned with the analysis of results and not the statistical analysis. Please retitle the section to "Analysis of results" and add the statistical methods used such as standard deviation etc.

Response: Kindly thanks for your reminding. “Statistical analysis” has been replaced with “Analysis of results” in the revised manuscript. Please refer to Line 184 in the revised manuscript. Statistical methods used such as standard deviation were added, and please refer to titles of Table 1, Figure 2 and Figure 3 in the revised manuscript.

  1. Table 1. is not very readable and needs to be rewritten or reoriented to horizontal.

Response: Thanks for your comments. Table 1 contains the data of residues, dissipation kinetics and residue accumulation (RA) of fluopyram and trifloxystrobin and their metabolites FLB and TRA in cucumbers and cowpeas after repeated spraying. Table 1 contains too much data. As you suggested, we have attempted to rewrite this table and reoriente it to horizontal. However, it is still not very readable and gets poor visual effects. Moreover, another Reviewer said that Figure 3 and Table 1 are very good. Therefore, we have only optimized this table on the basis of the original Table 1 to make it more readable. Please refer to Table 1 in the revised manuscript. If it still has any problem, please do not hesitate to contact us, and we will try our best to polish the manuscript. Thanks again for your good suggestion.

  1. Figure 2 and 3: Please explain in the figure caption what the error bars mean (It is standard deviation?)

Response: Thanks for the reviewer’s good question. The error bars represent the standard deviation. The “note” has been added in Figure 2 and 3 to explain the meaning of the error bars. Please refer to Figure 2 and 3 in the revised manuscript.

  1. In my opinion, the discussion part of the manuscript should be expanded. The obtained results should be discussed more with the results of other scientists.

Response: Special thanks for your good suggestion. We have compared our results with the results of other scientists in the discussion. Moreover, the corresponding references were added in the revised manuscript. Please refer to Line 356-359, 366-367, 376-383, 389-391, etc. in the revised manuscript.

  1. Conclusion: Please indicate whether the research objective was achieved or the research question was answered. Please suggest also what still needs to be studied in the future and emphasize why the research conducted is important compared to others.

Response: Thanks for your good question. The research objectives of the present study contained three aspects. Firstly, greenhouse-field trials were conducted to investigate the dissipation behavior and accumulation potential of pesticides applied to cucumbers and cowpeas. Secondly, a preliminary study of changes in pesticide residues in cucumbers and cowpeas was conducted via several traditional household processing operations. Thirdly, the health risks associated with exposure to pesticides were assessed on the basis of our residue data. The research questions were all answered in the Conclusions. To the best of our knowledge, this study constituted the first to systematically assess the dissipation, metabolism, accumulation, processing and risk assessment of fluopyram and trifloxystrobin in cucumbers and cowpeas from cultivation to consumption. Moreover, we also supplemented what still needs to be studied in the future in the Conclusions. Please refer to the Conclusions in the revised manuscript.

Reviewer 4 Report

The paper is well-written, contains clear and relevant information regarding the dissipation, metabolism, accumulation, processing and risk assessment of fluopyram and trifloxystrobin in cucumbers and cowpeas from cultivation to consumption. The introduction is brief and provides sufficient information on the importance of choosing the topic of this scientific paper.

The methods are generally appropriate and very well presented.

In the Results section, in table 1 and table 2 I suggest to analyse the differences between means and multiple comparison analysis using the t-test (two-sample assuming equal variances).

This work presents interesting results; the authors making an important contribution to the research literature in this area of investigation.

Author Response

Reviewer #4:

The paper is well-written, contains clear and relevant information regarding the dissipation, metabolism, accumulation, processing and risk assessment of fluopyram and trifloxystrobin in cucumbers and cowpeas from cultivation to consumption. The introduction is brief and provides sufficient information on the importance of choosing the topic of this scientific paper.

Response: Thank you very much for your comments and advice concerning our manuscript. Those comments are all valuable and helpful for revising and improving our manuscript. Based on these comments and suggestions, we have made careful modifications on the revised manuscript. If it still has any problem, please do not hesitate to contact us, and we will try our best to polish the manuscript.

  1. The methods are generally appropriate and very well presented.

Response: Thank you for your recognition of my work. We will keep working to improve our manuscript.

  1. In the Results section, in table 1 and table 2 I suggest to analyse the differences between means and multiple comparison analysis using the t-test (two-sample assuming equal variances).

Response: Thanks for the reviewer’s good question. In Table 1, we presented the residue data of fluopyram and trifloxystrobin and their metabolites fluopyram benzamide and trifloxystrobin acid in cucumbers and cowpeas at 2 h, 1 d, 3 d, 5 d, and 7 d after repeated spraying. These data were mainly used to calculate the dynamic equations, half-lives and esidue accumulation (RA) values. It's not really necessary to compare the residue differences between samples colleced from different time points. In Table 2, the national estimated daily intake (NEDI) and its corresponding chronic risk quotient (RQc) values were calculated on the basis of supervised trial median residue (STMR) values at the pre-harvest interval (PHI) day. We can only obtain a single figure for NEDI and RQc at each pre-harvest interval (PHI) day; and therefore, differences between means and multiple comparison analysis using the t-test were not necessary. Moreover, the international estimation of short term intake (IESTI) and its corresponding acute risk quotient (RQa) were similar as NEDI and RQc values. We acknowledge your questions are very valuable and helpful for our follow-up study, as well as important guiding significance to our researches. If it still has any problem, please do not hesitate to contact us, and we will try our best to polish the manuscript. Thanks again for your good suggestion.

  1. This work presents interesting results; the authors making an important contribution to the research literature in this area of investigation.

Response: Thank you for your recognition of my work. We will keep working to improve our manuscript.

Round 2

Reviewer 2 Report

Considering the changes proposed by the authors, I suggest the manuscript can be accepted in its present form. 

Reviewer 3 Report

Thank you for resposing to my comments.